# The Road Less Traveled: How COVID-19 Patients Use Metaphors to Frame Their Lived Experiences

**DOI:** 10.3390/ijerph192315979

**Published:** 2022-11-30

**Authors:** Yu Deng, Jixue Yang, Li Wang, Yaokai Chen

**Affiliations:** 1College of Language Intelligence, Sichuan International Studies University, Chongqing 400031, China; 2School of English Studies, Sichuan International Studies University, Chongqing 400031, China; 3Zengcheng Experimental School of Guangzhou Zhixin Middle School, Guangzhou 511300, China; 4Science and Education Department, Chongqing Public Health Medical Center, Chongqing 400036, China; 5Department of Infection Diseases, Chongqing Public Health Medical Center, Chongqing 400036, China

**Keywords:** metaphor, COVID-19 patients, mental health, narrative discourse, healthcare

## Abstract

Metaphor provides an important intellectual tool for communication about intense disease experiences. The present study aimed to investigate how COVID-19-infected persons metaphorically frame their lived experiences of COVID-19, and how the pandemic impacts on their mental health burden. In-depth semi-structured interviews were conducted with 33 patients afflicted with COVID-19. Metaphor analysis of patient narratives demonstrated that: (1) COVID-19 infection impacted patient conceptualization of themselves and the relationship between the “self” and the body, as well as social relationships. (2) Metaphors relating to physical experience, space and time, and integrative behaviors tended to be used by COVID-19 patients in a negative way, whereas war metaphors, family metaphors, temperature metaphors, and light metaphors were likely to express positive attitudes. (3) Patients preferred to employ conventional metaphors grounded on embodied sensorimotor experiences to conceptualize their extreme emotional experiences. This study has important implications with respect to the therapeutic function of metaphors in clinical communication between healthcare professionals and COVID-19 patients.

## 1. Introduction

Since the outbreak of COVID-19, a number of studies have investigated the physical health of COVID-19 patients, while studies of mental health problems relating to COVID-19 infection have been underestimated and poorly represented in the published literature. Previously published studies have shown that COVID-19 has an adverse impact on the mental health of COVID-19 patients [1,2,3,4,5], healthcare professionals [6,7,8], and the general public [9]. People in the general population have developed symptoms of anxiety, depression, and fear due to social isolation, lack of accurate and up-to-date scientific knowledge, and heavy workloads [10,11].

In the present COVID-19 era, language plays an important role in healthcare research as language can influence people’s experience of disease [12]. As a fundamental way of thinking, metaphors are widely used when people describe their personal experiences concerning social taboos and diseases [13,14,15]. When people experience a debilitating or painful disease, they employ metaphors to communicate details about their personal experience [12,13,15,16,17,18]. Currently, a growing body of published literature has shown that metaphors have been extensively used in communication regarding the COVID-19 pandemic, in political discourse [19,20,21,22], news discourse [23], and narrative discourse [24].

In the context of narrative discourse, previous research has focused on COVID-19 metaphors in online social media texts and posts made by the general public [25,26,27]. Wicke and Bolognesi [25] examined how people conceptualize COVID-19 using “war metaphors”, “monster metaphors”, “storm metaphors”, and “tsunami metaphors” in online Twitter writing. Analysis of data related to these metaphors used on social media showed that “war metaphors” accounted for the largest ratio among all types of metaphors. Semino [26] investigated metaphors concerning COVID-19 based on the large-scale online corpus. The research findings showed that “war metaphors” was frequently used to discuss COVID-19. Although “war metaphors” can have specific advantages in communication, their potential negative impact should not be underestimated. The function of metaphors should be adopted as the criterion to assess whether certain metaphors have positive or negative impacts. Interestingly, Semino [26] observed that the use of “fire metaphors” was more frequent compared to “war metaphors” in the framing of COVID-19. Tay [27] studied the factors impacting people’s preference for choosing “social unrest” or “COVID-19” as the source domain in constructing a pandemic-related metaphor, namely “COVID-19 IS THE SOCIAL UNREST” or “THE SOCIAL UNREST IS COVID-19”. Results indicate that people tend to use the more serious experience as the source domain. Specifically, those who regard social unrest as a more serious crisis than COVID-19 might prefer metaphors with social unrest as the source domain.

Recently, an increasing number of studies have addressed the mental health of COVID-19 patients through their lived experience narratives of COVID-19. Metaphors emerged in patient narratives regarding their unpleasant experiences during COVID-19 [1,2,24,28,29]. COVID-19 survivors summarized their experiences mainly using negative-oriented metaphors [28]. For example, COVID-19 patients expressed their pain with metaphors such as, “my kidneys were going to jump out of my body” [2]. Metaphors about quarantine and social isolation such as, “sometimes I felt like I am in jail”, were extensively used in patients who were infected for the first time [29]. COVID-19 patients who experienced re-test positivity considered their re-test positivity experiences as “life imprisonment”, or “a knot in the heart” [1]. To gain a deeper insight into how people conceptualize COVID-19 with metaphors, Gök and Kara [30] explored metaphors used by 210 people, including 17 COVID-19 patients. The study required participants to complete the following sentence: “COVID-19 pandemic is like... Because…”. Their results observed that “anxiety/concern” metaphors under the theme of “risk” were the most commonly used by COVID-19 patients. The findings revealed that the uncertainty related to COVID-19 increased the mental health burden of COVID-19 patients. It is noteworthy that patients not only spoke or communicated using metaphors but also “did metaphors or behaved in metaphorical ways” [15,31,32]. Along this line of inquiry, Deng et al. [24] employed semi-structured interviews to investigate how COVID-19 patients and healthy citizens from Wuhan used metaphors to frame their lived experience of COVID-19. Their results indicate that an extensive range of embodied metaphors (i.e., body parts, battling, hitting, weight, temperature, spatialization, motion, violence, light, and journeys) manifested in both verbal expressions and symbolic behaviors (e.g., COVID-19 patients drafted a will). These findings suggest that the physical manifestations of COVID-19, the environment, and co-existing psychological factors combine to intricately sculpt the metaphorical thought processes of COVID-19 patients.

Existing published literature related to COVID-19 metaphors is limited in several ways. Firstly, previous works largely concentrate on metaphors used by healthy participants, with only marginal documentation of metaphors used by COVID-19 patients. The in-depth relationship between metaphors and the mental health burden of COVID-19 patients has been underestimated. Secondly, previously published studies have paid attention to COVID-19-related metaphors seen in social media, news discourse, and political discourse. Comparatively little research has systematically examined metaphors in the lived experience narratives of COVID-19 patients. Thirdly, studies on COVID-19-associated metaphors have focused extensively on “violence-related metaphors” and “war-related metaphors”. However, the quest for a more positive and effective metaphorical framework has been lacking in healthcare communication. Lastly, COVID-19 metaphors in verbal modes have been extensively evaluated, while comparatively scant attention has been paid to metaphorical behaviors.

To gain a deeper insight into metaphors used by COVID-19 patients in describing their lived experience of COVID-19, the present study conducted semi-structured interviews with thirty-three COVID-19 patients to systematically analyze metaphors used by COVID-19 patients, and to reveal their underlying psychological status. The aim was to provide empirical data in seeking more positive and appropriate metaphorical patterns in reframing COVID-19 disease, and offering insights into possible psychological interventions for COVID-19-afflicted patients.

## 2. Material and Methods

### 2.1. Research Design

This study employed semi-structured telephone interviews with 33 Chinese COVID-19-infected persons from September 2020 to March 2021 to collect lived experience narratives [3]. Metaphor analysis was conducted with respect to the transcribed narrative data to reveal COVID-19 patients’ psychological status. The research team (two males and two females) encompassed one MD, one PhD, one Master of Medicine, and one Master of Art. Their occupations included doctor, university researcher, and graduate student. The researchers were experienced in qualitative research in medicine and social science.

### 2.2. Participants

Participants were recruited at the Chongqing Public Health Medical Center, China through convenience sampling. Five participants refused to take part in the interviews due to busy schedules and privacy concerns. Thirty-three patients with COVID-19, aged from 21 to 57 (mean age = 39), attended semi-structured interviews on the phone as paid volunteers. Eighteen participants were male and fifteen were female. The inclusion criteria were a confirmed diagnosis of COVID-19 and a presence of the symptoms of COVID-19 disease. Furthermore, all patients were admitted to the hospital for quarantine and treatment. Among the patients, six were admitted to the hospital more than once because of retest positivity subsequent to recovery. The mean hospitalization duration was 17 days. Table 1 summarizes the socio-demographic information of the 33 participants. We reached theoretical saturation by interviewing the 33 participants according to the entry criteria.

Written informed consent was obtained from all participants before the interview, and ethics committee approval was received for this study from the Ethics Committee of Chongqing Public Health Medical Center (Approval Date: 4 September 2020; Approval Number: 2020-048-02-KY).

### 2.3. Research Instrument

Metaphors are usually used when people describe abstract ideas, feelings, and special experiences that are personal and/or difficult to express [13,15,30]. To gather information related to metaphors used by participants, a question list of abstract ideas, feelings, and special experiences was designed for the semi-structured interviews, as shown in Table 2. The questions were used as a guideline in interviews. The order that the questions were to be asked by the interviewer was flexible, in deference to the specific situations of the individual patients. The interview questions, prompts, and guides were designed by the research team. The interview guide was pilot tested in our study of embodied metaphor in communication about the first wave of COVID-19 in Wuhan [24].

### 2.4. Interview Procedure

Before the interview, written informed consent was obtained from all participants. Semi-structured interviews were conducted by means of free conversation, by asking participants open-ended questions, and ‘how’ and ‘why’ questions, as in Table 2 [3,33]. The interview emphasized listening to the patients. Figure 1 illustrates the procedure of the interviews. Individual interviews with participants commenced in September 2020 and ended in March 2021. Each of the interviews, lasting between 30 to 60 min, was conducted by cellphone or WeChat phone, and was audio-recorded. Upon completion of each interview, online questionnaires were completed by participants in order to collect their demographic information.

### 2.5. Data Analysis

The audio recordings of 33 patients were transcribed by native Chinese speakers for data analysis. The procedure for metaphor identification and analysis is shown in Figure 2.

During data coding, we combined Pragglejaz Group’s MIP (metaphor identification procedure) [34], Cameron’s VIP (vehicle identification procedure) [35], and Steen et al.’s MIPVU (metaphor identification procedure VU university) [36] to identify metaphors. Firstly, we read the entire discourse and identified the focus, which may have been a lexical unit or a phrase, of the sentence. For example, the lexical unit “jackal” and the phrase “riding a roller coaster” are the focus of the sentences “COVID-19 virus is a jackal” and “Those who had experienced the relapse of the disease felt like riding a roller coaster”. We assessed whether the focus of the sentence was metaphorically used or not by comparing its contextual meaning to its dictionary meaning. Expressions lacking explicit metaphorical forms were identified according to Steen et al.’s MIPVU approach [36]. For instance, the sentence “Generally speaking, my emotion was heavy” was identified as a metaphor although it lacks the explicit form of a metaphor. The contextual meaning of the lexical unit “heavy” refers to the pressure of the speaker. The contextual meaning of “heavy” differs from its literal meaning.

Metaphors were also classified according to different semantic fields [15]. For instance, “I felt the pressure from my heart. It pressed me like a balloon and I am afraid that it will explode one day” was classified into the “pressure” metaphor and the “explosion” metaphor. Notably, metaphorical behaviors were included in this study. According to Littlemore and Turner [15], the actions that people express their emotions with in some pictures, symbols, or rituals can be seen as a category of metaphor referred to as “symbolic metaphorical enactment”. For example, one patient wrote a will to compensate for her loss of hope. This was an instance of metaphorical enactment in behavior.

The topics of metaphors were identified according to the context, including both physical scenarios (e.g., quarantine in hospital; symptoms of COVID-19) and mental dimensions (e.g., fear of death; emotional state of being infected). For instance, “They are angels, who bring us hope and warmth” indicated the topic of “image of medical staff”.

Two independent coders identified the metaphors, and disagreement was resolved by a third coder. All metaphors were translated from Chinese into English by the authors.

## 3. Results

### 3.1. Overall Distribution of Metaphors in COVID-19 Patients

Our study identified 406 instances of metaphors in the 33 COVID-19 patients, and 48 metaphorical categories, as shown in Table 3.

In Table 3, it can be seen that most metaphors tended to be used by COVID-19 patients to express negative associations. This might be owing to the fact that metaphors are used to describe things that are more negative, sensitive, emotional, and indescribable using language [37]. Metaphor categories such as “the machine”, “pushing and pulling”, and “fighting or battling, or struggling” show equal quanta of negative and positive polarity. The war metaphor, family metaphor, light metaphor, temperature metaphor, fairness metaphor, and animal metaphor were more likely to be used positively. For instance, patients used the family metaphor, such as “healthcare staff brought me the warmth of the family”, to show gratitude to healthcare professionals.

Among all metaphor categories, the image metaphor was the most frequently used metaphor type to describe patients’ perception of COVID-19 (e.g., COVID-19 virus is a wolf), the healthcare professionals (e.g., the doctors and nurses are angels), and the hospital (e.g., the hospital was a haunted house). Notably, the image metaphor, motion metaphor, integrative behavior metaphor, container metaphor, pressure metaphor, war metaphor, family metaphor, closeness and distance metaphor, life and death metaphor, and the darkness and light metaphor rank among the top ten metaphors in our analysis. These top ten metaphor categories are associated with human embodied sensorimotor experiences, suggesting that COVID-19 patients tend to use the conventional experience to frame their lived experience of COVID-19 [24].

Regarding the narrative topics, 54 topics were identified among the 406 instances of metaphors, with the top 10 metaphorical topics listed in Table 4.

Table 4 shows that COVID-19 patients tend to use metaphors to express their personal experiences, such as COVID-19 infection, quarantine in hospital, treatment in hospital, attitudes to death, relationships with the medical staff, as well as experiences of social stigma.

### 3.2. Typical Metaphor Use in COVID-19 Patient Narratives

#### 3.2.1. War Metaphors

Previously published studies have criticized “war metaphors”, given that war metaphors highlight terrible and life-or-death aspects, and may have an adverse impact on people in a disadvantaged situation [20,38,39]. However, Semino [26] showed that war metaphors may have a positive function in certain contexts. War metaphors can frame the war between patients and disease on the one hand, and they can frame the conflict between patients and healthcare professionals as well as the war between patients and their treatment, on the other [13,40].

War metaphors used by COVID-19 patients in the present study yielded contradictory findings. First, most war metaphors in this study were positive (16 out of 19) for COVID-19 patients. Second, instead of framing the war between patients and healthcare professionals or the war between patients and treatment, war metaphors in this study tended to frame the war between patients and COVID-19, as well as the war between healthcare professionals and COVID-19. COVID-19 patients preferred to align healthcare professionals and themselves together as the enemy of the virus, thus constructing an alliance of the power of patients and healthcare professionals acting together to fight COVID-19. Consider the war between patients and COVID-19 in the following examples:

(1) I had three big meals every day. I kept myself full. I encouraged myself that I didn’t feel fearful about the COVID-19 virus and I would certainly defeat it. (Patient 5, female)

(2) I firmly believed that I can defeat the COVID-19 disease. (Patient 9, male)

War metaphors seen in the above examples (1–2) frame patients’ perception of the relationship between the COVID-19 disease and themselves. Previous research demonstrated that war metaphors used by patients with cancer represent a sense of lack of agency and being out of control [41]. However, war metaphors in our study, as seen in examples 1 and 2 above, reinforce the sense that patients can control the war and express their strong will to recover from COVID-19 infection.

The following examples illustrate the war between healthcare professionals and COVID-19:

(3) They must be the fighters who fought against COVID-19 on the front line. This was a war. And we civilians were the ones who were protected by them. (Patient 33, male)

(4) They are like the heroes, the real heroes! They are not like the heroes, because they ARE the heroes. This was a war. This was a war without the smoke of gunpowder. (Patient 5, female)

(5) The doctors and the nurses were in the war. But we don’t have to be afraid because no matter who you are, you will receive the treatment if you have to. (Patient 12, female)

The preceding three examples illustrate the conflict between healthcare professionals and COVID-19, revealing patients’ praise and trust in healthcare professionals. Semino et al. [40] showed that violence-related metaphors have a negative function because these metaphors put patients in a position which is opposed to that of the disease, the treatment, and healthcare professionals. However, examples 3–5 demonstrate that patients saw healthcare professionals as the ones who were in conflict with COVID-19, and put themselves in a position of protection by healthcare professionals. Hence, the concepts of win-or-loss, vulnerability, and passivity related to war metaphors were not necessarily applicable to our observations. Instead, war metaphors in our study expressed the confidence of COVID-19 patients in winning the battle against COVID-19 with the help of healthcare professionals.

In the present study, war metaphors were likely to be used positively due to the influence of traditional Chinese culture, given that most offensive war metaphors used by Chinese people illustrate the conflict between the collective and the pandemic instead of the conflict between individuals (or the patient) and the disease. At the outbreak of a pandemic, war metaphors can contribute to encouraging people to confront the pandemic together, with unity [23].

#### 3.2.2. Journey Metaphors

Ten journey metaphors were identified in our data. Seven out of the ten journey metaphors were used negatively, to express the difficult or uncontrollable journeys patients went on, and these concur with the findings of Semino et al. [40]. Consider the following examples:

(6) When I tested positive for COVID-19 again after being cured, I was under extreme pressure. Because I didn’t know when the end of the journey was. I felt the greatest pressure when I read the news about reinfection. (Patient 7, female)

(7) I felt that this road was too difficult for me. It was completely different from the road I had been usually on. (Patient 10, male)

(8) For the whole process of our life, there are some difficulties, twists, and turn points in this process. (Patient 24, male)

In these examples, journey metaphors were used by patients to emphasize their helplessness due to COVID-19 infection. Patient 7 conceptualized the feeling of reinfection as an endless, arduous journey with enormous pressure. Patient 10 described his recovery process as a difficult road, which was very different from the common illnesses that he experienced in the past. Patient 24 expressed his ideas about disease as difficulties, twists, and turns in his life journey.

#### 3.2.3. Metaphorical Behaviors

Behaviors that people express their emotions with (e.g., rituals, symbols, or pictures) can be categorized into a category of metaphor called symbolic metaphorical enactment [15]. Metaphorical behaviors were observed in COVID-19 patient narratives. For example:

(9) I had an itch for pills, believing that I can only be cured by taking pills. I tried every means to beg the doctors to prescribe some pills for me. If they didn’t prescribe the pills for me, I thought that I would lose hope. But if they prescribed some pills for me, I thought that I was still filled with hope. (Patient 7, female)

Extract 9 is an example of symbolic metaphorical enactment that expresses the patient’s fear and worry. When the patient was admitted to hospital, she begged the doctors to prescribe medicines for her. In her mind, medicine was the only “symbol” that could help her. Medicines can be considered here as compensation for her loss of sense of security.

(10) I was so petrified that I even left a message; I mean the last words to my son, my parents, my brothers, and my sisters before death. (Patient 20, female)

In example 10, the patient was preoccupied with depression and fear due to her COVID-19 illness. During hospitalization, she chose to leave her last words to her family, which can be seen as a compensatory gesture for the loss of hope for life. This metaphorical behavior can confer an advantage in order for the patient to reconcile their negative emotional status [15,24]. For a patient who is approaching death, leaving some last words for family members is a physical ritual that can help to alleviate fear and depression at the time of potential end of life [15,24].

Jensen [31] proposed the notion of “metaphoricity”, which means that people not only “speak” metaphorically but also “do” metaphorically. Meaning potential is one of the most important features of the notion of “metaphoricity”. For example, “sit down” not only dictates motion but also has an implication of solving problems, which creates a metaphoric potential for “dissolving social conflict in sitting down together” [31] (p. 265). Consider the following metaphorical activities:

(11) When I was in a bad mood, I preferred to do the cleaning (Patient 4, male).

(12) One time when I took a walk at night, one of my neighbors ran to the right when he saw me walking on the left. He covered his mouth and nose and ran away. (Patient 18, female)

(13) Before I was infected with COVID-19, many young people would have a greeting with me when we met. But after knowing that I had been infected by COVID-19, they were afraid of me, keeping me far away. I was also filled with fear when I met some acquaintances. I knew that they petrified me so I consciously stayed away from them. (Patient 27, male)

Example 11 shows patient behavior in the face of social stigma. The action of “doing the cleaning” not only refers to the physical behavior of cleaning but also has an implication of struggling with negative emotions. Patient 4 tried to eliminate the negative emotion by doing the cleaning. This creates the metaphoric potential of “draining away the negative emotion is doing the cleaning”. Similarly, the metaphorical activities of intentionally keeping spatial distance (e.g., Patient 18 and Patient 27 in examples 12 and 13) embodied people’s stigma toward COVID-19 survivors, as if recovered COVID-19 patients remained carriers of the virus even though they had been cured and discharged from hospital. The potential meaning underlying the metaphorical activities has intensified the sense of social stigma and fear in Patient 18 and Patient 27.

#### 3.2.4. Physical Experience Metaphors

Metaphors of pressure, violence and impact, physical injury, divided self, as well as crumbling, breaking, and falling apart can be broadly included in physical experience-related metaphors.

Twenty pressure metaphors were observed in COVID-19 patients. Eighteen instances conveyed negative meaning, as described in the following extracts:

(14) I was depressed. I was pressed by a stone and I couldn’t push it away. (Patient 29, female)

(15) I was worried. I was very nervous like being pressed by a stone. It was very heavy. It was like a string that tightened me. I suffered a lot from COVID-19 disease because it compressed my mind and my spirit. It just felt like that. (Patient 5, female)

(16) When I waited for the result of the nucleic acid test, I felt that something was pressing against my heart. (Patient 30, female)

Examples 14 and 15 indicate the excessive pressure experienced by Patient 29 and Patient 5 due to COVID-19 symptoms. These patients felt helpless when infected with COVID-19. The infection was like an external force exerted on patients in a manner that was difficult to resist. These two metaphors express the disempowerment of the patients. Similarly, example 16 conceptualizes the unknown nucleic acid test result as an external force, revealing the extreme stress felt by Patient 30. Pressure metaphors in our findings were likely to be the target domain of “COVID-19 infection”. As a special and personal experience, COVID-19 infection caused agony in patients’ minds and evoked an abstract feeling that was difficult to express. Thus, “pressure” was framed as “CAUSE OF PHYSICAL DAMAGE” by patients when talking about the sense of psychological pain [42].

Metaphors of violence and impact in this study are closely associated with the embodied physical experience. These metaphors were mostly used negatively (seven out of ten), as illustrated in the following examples:

(17) I was dizzy, feeling that there was someone tying a rope around my neck and tightening it. I breathed hard, but I only took in very little oxygen. I felt like being beaten down by others and I would lose my attention. (Patient 1, female)

(18) We dare not tell our closest friends that we have been infected by COVID-19. If they know that, we won’t have friends anymore. To be honest, it is a great hit on us COVID-19 patients. (Patient 27, male)

(19) COVID-19 was trying to beat you down. It was trying to beat down your mind, your spirit and you. (Patient 5, female)

Examples 17–19 are typical violence-related metaphors. These three examples indicate the complex psychological scenarios present when patients lost control of their emotions due to infection or social stigma. Patient 1 was “beaten” by the symptoms of COVID-19, Patient 27 was “hit” by discrimination by others, and Patient 5 was “defeated” by the infection. Here, the extreme feelings and negative emotions of COVID-19 patients were metaphorically framed as physically “being beaten down” or “being hit”. These three metaphors were used in a disempowering manner, showing the helplessness and depressed feelings of these patients. The violence-related metaphors had a negative impact on COVID-19 patients, and thus should be avoided in healthcare communication with patients.

Metaphors of physical injury were used by patients to express the negative emotional feelings of patients in terms of the experience of injury to the human body, as shown in the following excerpt:

(20) The exposure of the patients’ personal information was inhuman. It was outrageous. Why did they rub the salt in our wounds? (Patient 3, female)

In this extract, Patient 3 discovered that her personal information was exposed by others and a seal was affixed to her front door by her neighbors. The COVID-19 infection aggrieved her, and the exposure of her personal information further increased her grief and sadness. The discrimination by her neighbors was likened to salt being rubbed in the wounds on her body. The source domain of this metaphor originates from physical injury in daily life, which can express the abstract inner feelings and pain of the patient.

“Divided self” metaphors express patient perception of the division between their bodies and the “self” [15]. In our study, six instances of metaphors relating to “divided self” were identified which expressed negative meaning. Consider the following examples:

(21) I couldn’t sleep at that time. I tried hard to relax. I tried hard to fall asleep. But my brain was out of control. It kept thinking. It kept entertaining foolish ideas. (Patient 10, male)

(22) I really wanted to distract my attention from the infection. I tried hard to think about other things. But my brain was out of my control. I kept thinking about the COVID-19 virus every day. It didn’t think of other things. It kept thinking about the virus. I tried hard to skip this thought. But I couldn’t. I couldn’t. (Patient 10, male)

In the two preceding examples, Patient 10 personified his brain and separated himself from it. In doing so, he blamed the symptoms of insomnia, anxiety, and daydreaming on his brain. This is a type of coping mechanism in which the patient blames illness on the body, which is separated from the “self” of the patient [15]. The “divided self” metaphors seen here emphasize the helplessness and lack-of-control present in the patient. The emotional negativity of this metaphor reflects patient disempowerment associated with an unexpected and serious infection [40].

Eleven metaphors of “crumbling, breaking, and falling apart”, used to express abstract feelings concerning COVID-19 infection in terms of physical experience, were observed in this study. Consider the following examples:

(23) At that moment, well, at the moment when I got to know the test result, I was insane. My mind has gone blank. I could hear nothing no matter who was talking to me. When I got to know the result, I felt that the sky collapsed. I was nervous. (Patient 4, male)

(24) I didn’t read any kinds of news. That’s important. Don’t read any news because the news could make you collapse. (Patient 28, male)

These above two extracts conceptualize the negative emotions of patients concerning COVID-19 disease in terms of the bodily experience of falling down and collapsing. Extract (23) shows the helplessness of Patient 4 when he was confirmed to be infected, and extract (24) demonstrates the fear in Patient 28 when he read news related to COVID-19. Metaphorically framing their negative feelings with the physical activity of “falling down” betrays the occult mental health burden of COVID-19 patients in their extreme situations.

#### 3.2.5. Space- and Time-Related Metaphors

Metaphors of closeness and distance, container, spatialization, and going back and forth are concerned with space and time. These metaphors were extensively used by patients to frame their emotions and feelings during COVID-19.

Fourteen instances of “closeness and distance” metaphors were used by patients to frame either the status of being close to death or the feeling of isolation from others. For example:

(25) If you were in my situation, you would have felt that death was really, really close to you. (Patient 21, male)

(26) He is a doctor of the department of infectious diseases, where we were quarantined. He was a million miles away from us. We didn’t have contact with him. (Patient 16, male)

(27) I considered that I was an unhealthy person. I wouldn’t like to stay with my family. I hoped that I could go to a distant place, keeping far away from them. I wanted to be with my family members after I was completely cured. I considered myself a person who might infect others. These thoughts always lingered in my brain. (Patient 7, female)

Extract 25 shows the patient’s fear of death in terms of proximity in space. Extract 26 describes the patient’s perception of social stigma originating from healthcare professionals in terms of extreme distance. In extract (27), Patient 7 desired to be isolated from her family because she worried about infecting family members. Long-term quarantine at the hospital aroused the sense of social isolation in COVID-19 patients, and these patients tended to metaphorically conceptualize their anxiety, depression, and stress by means of spatial distancing [24].

Twenty instances of container metaphors were used by patients (with fourteen negative instances). Container metaphors were likely to metaphorically represent the concept that they were confined to a specific spatial container due to the COVID-19 infection. Consider the following examples:

(28) I bore heavy mental pressure. According to the doctor, some people would stay as peacefully as they were in ordinary life. But I couldn’t walk out of it. (Patient 7, female)

(29) I was trapped in a predicament, being eager to escape. (Patient 1, female)

In extract 28, Patient 7 expressed the feeling of pressure and depression due to COVID-19 infection by conceptualizing her mental status as being in a container from which she could not escape. In extract 29, the confirmed COVID-19 status and the difficulties arising from infection was also expressed as being confined to a container, which metaphorically refers to the emotions of fear, worry, and depression in Patient 1. Our findings echo those of previous investigations, in that patients with depression tend to narrate that they fail to escape from the “container” of the depressed emotion [43].

It is noteworthy that patients in our study also conceptualized their body parts as a container, which was filled with the “substance” of abstract emotion. For instance:

(30) When they told me to transfer to another hospital, my heart was…well I didn’t know what had happened. When I was transferred to a higher-level hospital, my heart was without bottom. I was filled with fear. I was afraid that I would be an experimental article of the hospital. (Patient 10, male)

(31) They informed me that I was tested positive for COVID-19. When I got to know the news, my heart was lacking bottom. I was filled with tension and pressure. (Patient 7, female)

In extracts 29–30, Patient 10 and Patient 7 considered the “heart” organ as a container, and the substances inside the container were the emotions of fear and tension. When their hearts “lacked bottom”, their fear and pressure was overflowing out of the container like heated fluid.

Three instances of “spatialization metaphors” were used by patients in terms of spatial motion or containers, as narrated by Patient 29:

(32) Caught in COVID-19, I felt like taking a roller coaster. I fell down suddenly from the highest point to the lowest point. It was really like taking a roller coaster. I couldn’t stand it. (Patient 29, female)

In extract 32, Patient 29 conceptualized her experience of COVID-19 infection by means of the physical experience of riding a roller coaster, which activates the memory of tension and excitement. This example reflects the orientation metaphors, “HAPPY IS UP” and “SAD IS DOWN”. The infections of COVID-19 made patients feel “down”, which metaphorically reflects the symptoms of fear, anxiety, and stress in patients. Framing the psychological status of COVID-19 infection through a spatial experience of motion could assist with our deep understanding of the emotional status of frustrated patients.

The metaphor of “going back and forth” has a lasting implication for displacement in space and time, as described by Patient 10:

(33) In the beginning of COVID-19, I considered that I was in Beijing in 2002 with SARS. This COVID-19 might be the same as that. (Patient 10, male)

In extract 33, the experience of COVID-19 is being related to the 2002 outbreak of SARS. When patients mention the negative emotion regarding their past experience, psychotherapists are encouraged to lead the patients to re-experience what had happened before from a positive perspective, which can provide a further psychological intervention for the patients with negative emotions in the present [44].

#### 3.2.6. Image Metaphors

Image metaphors, which were used to express patients’ self-conception or social identity, accounted for the highest proportion of metaphors. Patients employed image metaphors to describe the change in their identity because the disease of COVID-19 had affected their concept of themselves. They considered themselves as “the prisoner who was incarcerated”, as shown in the following examples:

(34) I felt like a prisoner, and I was serving a life sentence. I didn’t know when I can go back home. (Patient 7, female)

(35) We are victims, not prisoners. Sometimes I encountered discrimination from others. But we are not prisoners. We are victims. (Patient 24, male)

Long-term quarantine brought Patient 7 and Patient 24 the strong sense of being a “prisoner”. As a victim of COVID-19 infection, Patient 7 highlighted that she had no freedom, and Patient 24 emphasized that he was innocent. The image of being a “prisoner” has ruined the self-esteem of patients, and has augmented their negative emotions [1,2,10,11,45].

The image metaphor can be used by patients to express their identity, as described by Patient 20:

(36) That’s right, that’s right. Those who were not cured were still in quarantine. I admitted that we were offered some good meals but I felt like I was a prisoner who had committed a crime. (Patient 20, female)

(37) At that time, I considered myself a plague. I used to be afraid of the plague but at that time I suddenly realized that I was the plague. For the first time, I realized that I WAS the plague. (Patient 20, female)

Extract 36 concerns the stigmatization of patients, given that COVID-19 patients were seen as prisoners who had committed a crime. This negative image has made patients feel angry, anxious, and fearful. In extract 37, the “plague” was used by Patient 20 to illustrate the sense of self-stigmatization and shame, suggesting that COVID-19 had an adverse effect on patients’ self-conceptualization and their identities. In the present era of COVID-19, negative image metaphors such as “prisoner” and “plague” should be strongly and emphatically discouraged in healthcare communication.

#### 3.2.7. Positive Metaphorical Framing

In the present study, the family metaphor, light metaphor, temperature metaphor, and most war metaphors were used in a positive way. These metaphors are encouraged in healthcare communication and therapy.

Family metaphors were used to narrate patients’ attitude towards or conception of healthcare professionals, volunteers, and the country. These metaphors show positive emotions such as love and praise, as narrated by Patient 16:

(38) I am so proud that I was born in China, which is a big family. (Patient 16, male)

Family metaphors can bridge the relationship between COVID-19 patients and the country, the healthcare professionals, and the volunteers. Family metaphors are beneficial in encouraging people to collectively fight against the COVID-19 pandemic.

The light metaphor had a positive effect on COVID-19 patients, as described by Patient 2:

(39) Healthcare staff is the sun that is shining with boundless radiance. Every time they came in, they brought us light and hope. (Patient 2, female)

As shown in extract 39, Patient 2 describes the pandemic as the darkness, and conceptualizes healthcare professionals as the light that can disperse the darkness of COVID-19. The use of light metaphors encourages patient motivation and hope. Furthermore, the metaphorical framing of the conflict between light and darkness can result in avoidance of the conflict between the individual and the COVID-19 pandemic.

Warmth metaphors were used positively by COVID-19 patients, as narrated by Patient 11:

(40) The strength of our country and the care and treatment from the doctors and nurses in the hospital warmed me a lot. They did not take me as a patient but made me feel the warmth of the family. (Patient 11, male)

In extract 40, both the warmth metaphor and the family metaphor were used by Patient 11 to express positive emotion and feeling in the recovery process, with adequate support from the government and medical staff.

## 4. Discussion

The purpose of this study was to explore the emotional states and the psychological milieu of COVID-19-afflicted patients through metaphor analysis identifiable in narratives of their lived experiences of COVID-19. Metaphor analysis demonstrates that COVID-19 impacted on social relationships of patients, self-conceptualization, and the relationship between “selves” and physical bodies. Metaphors relating to physical experience, space and time, and integrative behaviors tended to be used by COVID-19 patients in a negative way, whereas the war metaphor, family metaphor, temperature metaphor, and light metaphor were likely to express positive attitudes. The analysis of metaphors in COVID-19 patient narratives may offer clinical correlates for utilization of literal and metaphorical meanings in pandemic-related healthcare communication [24].

In the literature, the extensive use of war metaphors in disease discourse, such as that around COVID-19, has been criticized for its potential negative implications in heath communication [19,20,21,22,25,26,38,39]. The present study, however, has a conflicting observation to make, in that war metaphors used by COVID-19 patients in China are used to show a positive attitude towards the struggle against COVID-19, while journey metaphors were employed in a disempowering way. This is consistent with Semino’s [26] corpus-based study on the emotionality of war metaphors in online texts. Therefore, whether the potential meaning of a metaphor is positive or negative should be inferred according to its function in the communicative context rather than their metaphoric category [26,40]. In our interviews, war metaphors tended to be used by COVID-19 patients to describe the conflict between the patients themselves and COVID-19 or between healthcare professionals and COVID-19. In the clinical communication between healthcare professionals and COVID-19 patients, war metaphors should be appropriately used [26]. The present study promotes the usage of war metaphors in framing the conflict between the collective and the pandemic instead of the “war” between the individual (i.e., the patient) and the disease [23]. In other words, cooperation and unity should be emphasized and reinforced when war metaphors are employed by healthcare professionals during health communication with COVID-19 patients.

With regards to metaphorical behaviors, symbols and rituals can offer insights into the assessment of mental health of patients with COVID-19 [24]. In the present study, integrative behaviors, such as the drafting of a will, the obsessive behavior of doing the cleaning, and begging doctors for medicine reflect COVID-19 patients’ symptoms of anxiety, stress, and depression when they were confirmed with a diagnosis of COVID-19. This is consistent with our assumption that metaphorical symbolic behaviors can reveal the extended emotions of the lived experiences of COVID-19 patients during COVID-19 infection [24]. In clinical practice, healthcare staff should pay mindful attention to both their language and behaviors when treating patients with COVID-19. Intentionally or unintentionally keeping away from patients is likely to compound their mental health burdens [1].

In the present study, metaphors relating to pressure, violence and impact, physical injury, and the divided self were all identified as physical experience metaphors. These metaphors were conventional, and originate from ordinary embodied experiences [13,24]. Patients suffering from COVID-19 disease are very likely to use metaphors relating to physical experience in a negative manner [24]. Thus, the pressure metaphor, violence and impact metaphor, physical injury metaphor, and the divided-self metaphor should be used with caution in healthcare communication. When these metaphors are used by patients to express their negative emotions, therapists can utilize the “metaphor-body-psychotherapy” model [46] to conduct body-based interventions for patients based on the embodied motor-sensory experience. For example, when COVID-19 patients narrate that “the exposure of my personal information was rubbing the salt in my wound” (extract 20), healthcare professionals can use positive body metaphors, such as “imagining the fast recovery of a cut scar”, to guide patients to “repair” the physical wound in order to heal the psychological trauma.

Space- and time-related metaphors, including metaphors of “closeness and distance”, “container”, “spatialization”, and “going back and forth”, were used by patients to express their complex emotions and psychological status due to COVID-19 infection [24]. Closeness metaphors were used when patients talked about death. COVID-19 disease and symptoms such as breathing difficulties “bridged” the relationship between patients and death. The painful experiences resulted in patients’ perception of being close to death. Distance metaphors were employed by patients to express their isolation from normal society and other people. Our findings reveal that quarantine-related social isolation was a major risk factor contributing to psychological problems in COVID-19 patients [1,2,3,4,28]. Furthermore, the fear of infecting others had an adverse impact on the psychological status of patients [3,29]. Patients in the present study deliberately kept away from family members because they considered themselves as unhealthy persons carrying the virus, in spite of recovery or hospital discharge. Similar findings have been documented recently in our qualitative emotional analysis of COVID-19 patients in China [3]. Therefore, adequate social support and accurate scientific knowledge regarding COVID-19 are urgently needed for patients [1,2,3,4,11]. Container metaphors were used by COVID-19 patients to either conceptualize their bodies being full of negative emotions such as depression, anxiety, and fear, or to frame their experience of being trapped in an isolated space such as in the hospital. In these scenarios, psychotherapists or healthcare professionals should encourage patients to speak about the shame, or the tension, fear, and pressure hidden in the “black box” of their minds. This describes the “problem-solution framework” with the “container metaphor” [46]. The metaphor of “going back and forth” can refer to the notion of “Psychological Transference”, whose source domain and target domain are “present” and “past” [44]. This category of metaphor is aligned with the underlying psychological processes inherent to COVID-19 patients, in that it can trigger patients to re-experience past events. In clinical intervention, helping patients to re-experience past events positively can exert a positive effect on the present treatment of negative emotions in these patients [44].

Negative image metaphors (e.g., plague, prisoner) were frequently used by COVID-19 patients to describe their concept of selves, suggesting that metaphors play an important role for patients to construct their identities [47,48]. This echoes Palese et al.’s findings in that the negative-oriented metaphors used by COVID-19 patients revealed their increased vulnerability due to stigmatization and socioeconomic difficulties and the limited support received [28]. Healthcare professionals should employ positive image metaphors such as “the fighter” or “winner” in communication with COVID-19 patients in order to help them counter social stigma and to reconstruct their identities [16,24]. In addition to image metaphors, the present study showed that family metaphors, light metaphors, warmth metaphors, and some war metaphors had a positive effect on patients with COVID-19. This is consistent with the recent literature [3,28] in that COVID-19 infection has provided a special occasion for meaning and for turning the life-threating event into a positive opportunity in fighting the disease and strengthening social relationships. Therefore, these positive metaphorical frameworks can be extensively applied to psychotherapeutic principles used in the treatment of COVID-19. Overall, our findings corroborated the observations of previous literature, in that we should decide the positivity or negativity of metaphors according to their functions in specific communicative contexts rather than their categories [40]. Hence, healthcare professionals should decide whether specific metaphors are useful or not according to the specific situation when communicating with patients.

### Limitations and Future Research

This study had some limitations. First, data collection was restricted to one institute. The results may not be generalizable to a wider population. Second, interview transcripts and research findings were not returned to participants for comments. In light of the potential challenges, future research could conduct interviews with participants from different institutes around the whole country. Furthermore, the research findings regarding metaphorical thinking of COVID-19 infection can be shared with participants. This can help participants understand the mental and physical status of other COVID-19 patients.

## 5. Conclusions

The present study observed that most metaphors in COVID-19 patient narratives were used negatively, except for family metaphors, warmth metaphors, light metaphors, and some war metaphors. Patients’ negative metaphorical framings of COVID-19, such as metaphors of pressure, violence and impact, physical injury, and the divided self, reveal their mental health burden caused by long-term quarantine, social stigma, and fear of infecting others. This suggests that positive metaphors should be promoted and the use of negative metaphors should be avoided in healthcare communication. In addition, our findings show that metaphors may be expressed as words, but may also be expressed as specific behaviors. Metaphorical behaviors have the capacity to reveal the mental health status of patients with extremely negative emotions. Some unconscious metaphorical behaviors (e.g., drafting a will) could potentially cause damage to the psychological status of COVID-19 patients. Hence, mindful attention should be paid to both language and behaviors used during interactions with COVID-19 patients.

### Recommendations

Metaphors are helpful in healthcare communication, conferring advantages not only to the narration of complex emotional personal experiences, but also to the reconstruction of patient identity [15,42,47,48]. In the practice of pandemic-related psychotherapy, therapists may discover patient metaphors with negative, disempowering effects correlating with patient lived experiences, and may then adopt diverse metaphorical frameworks (i.e., verbal and behavioral metaphors) in positive, empowering ways to provide metaphor-based psychological interventions (e.g., metaphor-body-psychotherapy [46,49]). The ultimate purpose of this is to project positive emotion and attitudes towards COVID-19 and the patients afflicted with COVID-19, and to reduce the underlying mental health burden of COVID-19 patients during their recovery process.

## Figures and Tables

**Figure 1 ijerph-19-15979-f001:**
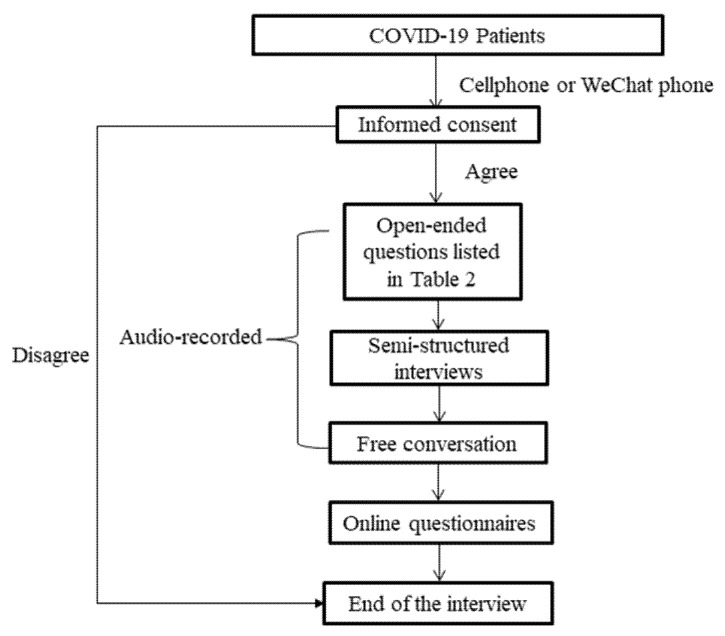
The procedures followed at each interview.

**Figure 2 ijerph-19-15979-f002:**
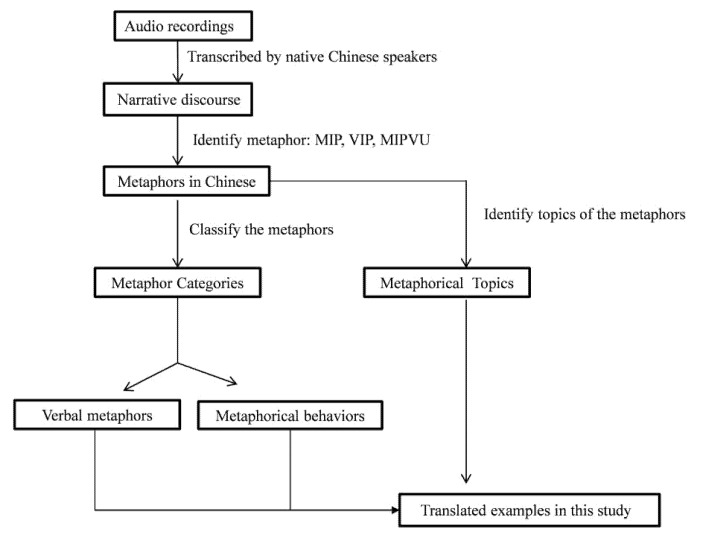
The procedure for metaphor identification and analysis.

**Table 1 ijerph-19-15979-t001:** Socio-demographic characteristics of COVID-19 patients (N = 33).

Characteristics	Number (%)
Gender	
Male	18 (54.5%)
Female	15 (44.5%)
Age	
20–30	8 (24.2%)
31–40	11 (33.4%)
41–50	8 (24.2%)
51–60	6 (18.2%)
Occupation	
Engineer	5 (15.2%)
Individual business	5 (15.2%)
Factory worker	4 (12.12%)
Farmer	4 (12.12%)
Unemployed	4 (12.12%)
Waiter	2 (6.1%)
College student	2 (6.1%)
Manager	1 (3.0%)
Medical staff	1 (3.0%)
E-commerce	1 (3.0%)
Housewife	1 (3.0%)
Retired	1 (3.0%)
Finance staff	1 (3.0%)
Teacher	1 (3.0%)
Marital status	
Married	24 (72.7%)
Unmarried	7 (21.2%)
Divorced	2 (6.1%)

**Table 2 ijerph-19-15979-t002:** Question list for the interview.

Scenarios	Interview Questions
Infection	How were you infected with COVID-19? How did you feel when you learned that you were infected?
Confirmation of Infection	Please say something about the whole process of going to the hospital, including the conditions on the way to the hospital, the conditions in the hospital, and the experiences of doing the nucleic acid testing? How did you feel when you learned the test result?
Unforgettable Events in Hospital	Please describe your treatment during your hospitalization. How did you feel in the hospital? What is your most unforgettable experience in the hospital?
Social Interaction	How did the doctors and nurses help you? Please describe an unforgettable story that happened between you and them.
Physical Symptoms	What symptoms did you have after you were infected? Was your condition serious? How did you feel when your condition got worse?
Mental Status	Please describe the process of your mental status during the hospitalization.
Thoughts during Isolation	What did you usually think about during the hospitalization? When did you feel unhappy after the infection?
Contagion	Have you infected others?
Hospital Discharge	Please describe the day when you were discharged from the hospital?
Social Stigma	Are you willing to talk about your COVID-19 infection experience with others? How do you get along with your family, relatives, friends or neighbors now? What is your attitude towards those who keep away from you?
Dreams	Please describe dreams that you had during the pandemic.
Future Life and Death	What do you think about your future now? What do you think about life and death? What is your attitude toward death after the COVID-19 infection?
The Image of COVID-19	If you have to draw a picture named COVID-19, what will you draw? Please describe the picture in your mind when talking about COVID-19.
The Country	Please describe your opinions on our country’s efforts to fight against COVID-19; Please describe the images of our country in your mind.
Healthcare Professionals	What is your impression of the healthcare professionals during COVID-19? Why?

**Table 3 ijerph-19-15979-t003:** Categories of metaphors among COVID-19 patients.

Metaphor Categories	Instances	Negative	Metaphor Categories	Instances	Negative
image metaphor	119	69	machine	5	2
motion	32	26	weight	5	4
integrative behavior	31	29	games, chance, sport	4	4
container	20	14	sense of touch	4	3
pressure	20	18	opening and closing	4	1
war	19	3	body metaphor	4	4
family	17	0	explosion	4	4
closeness and distance	14	13	physical injury	3	3
life and death	14	6	spatialization	3	2
darkness and light	12	3	up and down	3	1
animacy	11	7	presence and absence	3	2
crumbling, breaking, falling apart	11	11	hiding	3	2
violence and impact	10	7	agency, lack of agency	2	2
journey	10	7	liquid metaphor	2	2
temperature	8	0	pushing and pulling	2	1
color	7	5	inside and outside	2	2
size	7	2	going back and forth	2	2
different realities	7	5	depth	2	2
symbolic metaphorical enactment	6	5	fighting, battling or struggling	2	1
fairness, justice	6	1	finding and losing	1	1
animal	6	2	allusion	1	0
divided self	6	6	cleanliness, dirtiness	1	1
carrying	6	4	trials, law	1	1
conduit metaphor	5	3	balance	1	0

(Note: one instance of metaphor can belong to multiple categories).

**Table 4 ijerph-19-15979-t004:** Top 10 topics of metaphors seen in COVID-19 patients.

Topics of Metaphors	Frequency (Ratio)
1. Medical staff	44 (10.8%)
2. Quarantine in hospital	38 (9.4%)
3. Life and death (e.g., worry about death)	38 (9.4%)
4. Emotional state of being infected (e.g., worry, fear, anger, anxiety, panic)	35 (8.6%)
5. The country	32 (7.9%)
6. Image of COVID-19	19 (4.7%)
7. Discrimination	18 (4.4%)
8. Emotional state in the hospital	17 (4.2%)
9. Medical examination in hospital (e.g., nucleic acid testing)	16 (3.9%)
10. COVID-19 infection	12 (3.0%)

## Data Availability

The data presented in this study are available on request from the first author or the corresponding author. The data are not publicly available due to privacy reasons.

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
