# Peer review of "The Road Less Traveled: How COVID-19 Patients Use Metaphors to Frame Their Lived Experiences"

_ijerph, 2022, doi:10.3390/ijerph192315979_

Round 1
Reviewer 1 Report
Should include the females in gender, since in the results reference females patients responses beginning on line 183 pronoun "her".
Author Response
Dear Reviewer,
We appreciate the interest that you have taken in our manuscript. The comments are very insightful for improving our manuscript. We have made revisions in accordance with your suggestions. Revised and rewritten portions were marked in blue color in the revised manuscript. The point-by-point response to your comments is shown below:
Should include the females in gender, since in the results reference females patients responses beginning on line 183 pronoun "her".
Response: Thanks for the reminder. We have added the female information in Table 1 and the relevant gender descriptions “Eighteen participants were male and fifteen were female.” We have also added the gender information of participants for all the 40 examples.
Thank you for again for your interest and efforts and we are ready to respond to any further questions and comments you may have.
Best wishes,
The authors
Reviewer 2 Report
Dear Authors
Thank you for allowing me to review this manuscript. This is a qualitative study on how COVID-19 patients use metaphors to plan their experiences.
This is an interesting work that requires some relevant improvements and a thorough review by the authors before its publication.
-As the COREQ (Consolidated criteria for reporting qualitative studies) recommendations show, the Research team and reflexivity domain is important in any qualitative study. This aspect is not sufficiently addressed in your manuscript. They should include more data on the researchers (credentials, gender, occupation and experience in qualitative research). 
-Regarding sampling, authors should provide information on whether there were subjects who refused to take part and if so, the number and causes.
-Did participants provide feedback on the findings? Were transcripts returned to participants for comment and/or correction?
These aspects should be explained and included if it made them.
Author Response
Dear Reviewer,
We appreciate the interest that you have taken in our manuscript. The comments are very insightful for improving our manuscript. We have made revisions in accordance with your enlightening suggestions. Revised and rewritten portions were marked in blue color in the revised manuscript. The point-by-point response to your comments is shown below.
Reviewer 2
Dear Authors
Thank you for allowing me to review this manuscript. This is a qualitative study on how COVID-19 patients use metaphors to plan their experiences. This is an interesting work that requires some relevant improvements and a thorough review by the authors before its publication.
Response: We are grateful to the reviewer for the positive comments.
-As the COREQ (Consolidated criteria for reporting qualitative studies) recommendations show, the Research team and reflexivity domain is important in any qualitative study. This aspect is not sufficiently addressed in your manuscript. They should include more data on the researchers (credentials, gender, occupation and experience in qualitative research). 
Response: We have added the researcher information on page 3 (2.1. Research Design), as copied below:
“The research team (two males and two females) encompassed one MD, one PhD, one Master of Medicine, and one Master of Art. Their occupations included doctor, university researcher, and master student. The researchers were experienced in qualitative research in medicine and social science.”
-Regarding sampling, authors should provide information on whether there were subjects who refused to take part and if so, the number and causes.
Response: we have added the subject information on page 3 (2.1. Participants), as copied below:
“Participants were recruited at Chongqing Public Health Medical Center, China through convenience sampling. Five participants refused to take part in the interviews due to busy schedule and privacy concern. Thirty-three patients with COVID-19, aged from 21 to 57 (mean age=39), attended semi-structured interviews on the phone as paid volunteers.”
-Did participants provide feedback on the findings? Were transcripts returned to participants for comment and/or correction? These aspects should be explained and included if it made them.
Response: Sorry, participants did not provide feedback on the findings and transcripts were not returned to participants for comment and/or correction. We have admitted this limitation in the newly added section “Limitations and Recommendations” one page 17, as copied below:
“This study had some limitations. ……Second, interview transcripts and research findings were not returned to participants for comments. In light of the potential challenges……the research findings regarding metaphorical thinking of COVID-19 infection can be shared with participants. This can help participants understand mental and physical status of other COVID-19 patients.”
Thank you for again for your interest and efforts and we are ready to respond to any further questions and comments you may have.
Best wishes,
The authors
Reviewer 3 Report
The article written by Yu Deng et al. is interesting. There are few comments and suggestions that authors should consider to improve quality of manuscript further.
The authors should either use “SARS-CoV-2” or “COVID-19” and it is should be uniform throughout the manuscript.
Besides, authors should describe “SARS-CoV-2” or “COVID-19” in full form at first.
The authors should provide latest facts about COVID-19 in China. If possible, I would like to hear why China is still under lockdown despite they were in lead during the first wave of COVID-19 and world learned a lot from China. What could be the possible reasons of this?
Why this study was restricted to one institute only? This could have a significant impact on study findings. There will be issues with the generalizability of the findings of this work.
What do you mean by e clinical manifestations of COVID-19?
How the validation of the interview guide was ensured?
Who translated the Chinese to English?
How have you worked on saturation point?
The quotes of the participants based on the themes should be provided in a table.
The discussion section should be updated with the more recent studies.
Conclusion section needs a complete revision. It should be very precise based on your study findings.
Add recommendations of your study as a separate heading after conclusion.
Author Response
Dear Reviewer,
We appreciate the interest that you have taken in our manuscript. The comments are very insightful for improving our manuscript. We have made revisions in accordance with your enlightening suggestions. Revised and rewritten portions were marked in blue color in the revised manuscript. The point-by-point response to your comments is shown below.
Reviewer 3
The article written by Yu Deng et al. is interesting. There are few comments and suggestions that authors should consider to improve quality of manuscript further.
The authors should either use “SARS-CoV-2” or “COVID-19” and it should be uniform throughout the manuscript. Besides, authors should describe “SARS-CoV-2” or “COVID-19” in full form at first.
Response: We have used “COVID-19” throughout the manuscript as the uniform. In the first line of Introduction, we have added the full name of COVID-19 (Corona Virus Disease 2019)
The authors should provide latest facts about COVID-19 in China. If possible, I would like to hear why China is still under lockdown despite they were in lead during the first wave of COVID-19 and world learned a lot from China. What could be the possible reasons of this?
Response: The Chinese government has adopted the dynamic zero-COVID policy to face the challenge of the COVID-19 pandemic in China. The dynamic zero-COVID policy does not mean 'zero infection,' as currently it is impossible to ensure that not one single local case occurs within the country. Instead, it refers to a slew of measures to swiftly eradicate new outbreaks when they happen. The gist of the policy is to eliminate the spread of new infections as soon as is possible, instead of just letting go of the situation and allowing it to get out of control. According to Liang Wannian, head of the expert team on COVID-19 response and disposal at the National Health Commission (NHC), the dynamic zero-COVID policy is the most cost-effective way of combating the pandemic in China, and this approach has now been vindicated and has gained global recognition for saving human lives.
Admittedly, in order to implement the policy, small-scale lockdowns and domestic travel restrictions will necessarily need to be enforced for short periods of time in certain areas where new COVID cases have been detected. However, for the country as a whole, China is not under lockdown restrictions anymore, and has not been for some time. In fact, some of the travel restrictions to and from overseas that had been in place have recently been revoked by the government, and the country has thus opened itself more widely to the rest of the world.
Why this study was restricted to one institute only? This could have a significant impact on study findings. There will be issues with the generalizability of the findings of this work.
Response: We admit this limitation in the newly added section “Limitations and Recommendations” on page 17, as copied below:
“This study had some limitations. First, data collection was restricted to one institute. The results may not be generalizable to a wider population……. In light of the potential challenges, future research could conduct interviews with participants from different institutes around the whole country.”
What do you mean by the clinical manifestations of COVID-19?
Response: Sorry for the vague description. We have changed this wording into “clinical symptoms of COVID-19 disease” on page 3 (section 2.1. Participants), as copied below:
“The inclusion criteria were a confirmed diagnosis of COVID-19 and a presence of the clinical symptoms of COVID-19 disease.”
How the validation of the interview guide was ensured?
Response: we have added the validation of the interview guide in section 2.3. Research Instrument, on page 4, as copied below:
“The interview questions, prompts, and guides were designed by the research team. The interview guide was pilot tested in our study of embodied metaphor in communication about the first wave of COVID-19 in Wuhan [24].”
[24] Deng, Y.; Yang, J.; Wan, W. Embodied metaphor in communication about lived experiences of the COVID-19 pandemic in Wuhan, China. PloS One 2021, 16(12), e0261968.
Who translated the Chinese to English?
Response: The second author majoring in English translated the Chinese examples to English. The other three authors revised them. We have added one line at the end of section 2.5 on page 6:
“All metaphors were translated from Chinese into English by the authors.”
How have you worked on saturation point?
Response: We added the information in section 2.1. on page 3, as copied below:
“We reached theoretical saturation by interviewing the 33 participants according to the entry criteria.”
The quotes of the participants based on the themes should be provided in a table.
Response: we currently put the 40 extract quotes within the lines in Result descriptions according to the template of the journal. If the editorial office requires all the examples in a single Table, we are sure to provide all quotes of patients in a table as a supporting document or appendix.
The discussion section should be updated with the more recent studies.
Response: Thanks for this constructive comment. We have updated the discussion section with the more recent literature. For instance:
“The present study, however, has a conflicting observation to make……This is consistent with Semino’s [26] corpus-based study on the emotionality of war metaphors in online texts.” (see page 15)
“Patients in the present study deliberately kept away from family members……Similar findings have been documented recently in our qualitative emotional analysis of COVID-19 patients in China [3]. ”(see page 16)
“Negative image metaphors (e.g., plague, prisoner) were frequently used by COVID-19 patients……This echoes Palese et al.’s findings in that the negative-oriented metaphors used by COVID-19 patients revealed their increased vulnerability due to stigmatization and socioeconomic difficulties and the limited support received [28]. ”(see page 17)
“The present study showed that family metaphors, light metaphors, warmth metaphors, and some war metaphors had a positive effect…….This is consistent with the recent literature [3, 28] in that COVID-19 infection has provided a special occasion for meaning and for turning the life-threating event into a positive opportunity in fighting the disease and strengthening social relationships.” (see page 17)
Conclusion section needs a complete revision. It should be very precise based on your study findings. Add recommendations of your study as a separate heading after conclusion.
Response: we have revised the conclusion and cut it into one single paragraph. We added a separate heading after conclusion, namely “limitations and recommendations”. See pages 17-18, as copied below:
- 5. Conclusions
“The present study observed that most metaphors in COVID-19 patient narratives were used negatively, except for family metaphors, warmth metaphors, light metaphors, and some war metaphors. Patients’ negative metaphorical framings of COVID-19, such as metaphors of pressure, violence and impact, physical injury, and the divided self, reveal their mental health burden caused by long-term quarantine, social stigma, and fear of infecting others. This suggests that positive metaphors should be promoted and the use of negative metaphors should be avoided in healthcare communication. In addition, our findings show that metaphors may be expressed as words, but may also be expressed as specific behaviors. Metaphorical behaviors have the capacity to reveal the mental health status of patients with extremely negative emotions. Some unconscious metaphorical behaviors (e.g., drafting a will) can potentially cause damage to the psychological status of COVID-19 patients. Hence, mindful attention should be paid to both language and behaviors used during interactions with COVID-19 patients.”
Limitations and Recommendations
“This study had some limitations. First, data collection was restricted to one institute. The results may not be generalizable to a wider population. Second, interview transcripts and research findings were not returned to participants for comments. In light of the potential challenges, future research could conduct interviews with participants from different institutes around the whole country. Furthermore, the research findings regarding metaphorical thinking of COVID-19 infection can be shared with participants. This can help participants understand mental and physical status of other COVID-19 patients.
Metaphors are helpful in healthcare communication, conferring advantages not only to narration of complex emotional personal experiences, but also to the reconstruction of patient identity [15, 42, 47-48]. In the practice of pandemic-related psychotherapy, therapists may discover patient metaphors with negative, disempowering effects correlating with patient lived experiences, and may then adopt diverse metaphorical frameworks (i.e., verbal and behavioral metaphors) in positive, empowering ways to provide metaphor-based psychological interventions (e.g., metaphor-body-psychotherapy [46, 49]). The ultimate purpose of this is to project positive emotion and attitudes towards COVID-19 and the patients afflicted by COVID-19, and to reduce the underlying mental health burden of COVID-19 patients during their recovery process.”
Thank you for again for your interest and efforts and we are ready to respond to any further questions and comments you may have.
Best wishes,
The authors
Round 2
Reviewer 3 Report
The authors have addressed most of my concerns nonetheless, few changes are required to be made before formal acceptance this manuscript.
What is the difference between a symptom and clinical symptom? Please use this term carefully with a reference from the literature. This is still confusing.
The section of “Limitation” should be added before conclusion after discussion. Please make changes accordingly.
Author Response
Dear Reviewer,
We are very grateful to your 2nd round suggestions. We have revised accordingly and the revised portions were marked in“Track Changes” function. The point-by-point response to your comments is shown below.
The authors have addressed most of my concerns nonetheless, few changes are required to be made before formal acceptance this manuscript.
What is the difference between a symptom and clinical symptom? Please use this term carefully with a reference from the literature. This is still confusing.
Response: Thanks for the reminder. We have changed the term into “symptom of covid-19 disease” according to most published literature.
The section of “Limitation” should be added before conclusion after discussion. Please make changes accordingly.
Response: We have cut the “Limitation” to the end of Discussion, before Conclusion.
Thank you for again for your help and we are ready to respond to any further comments you may have.
Best wishes,
The authors